# Our good neighbors: Understanding ecosystem services provided by insectivorous bats in Rwanda

Olivier Nsengimana[1]☯, Faith M. Walker[2,3]☯*, Paul W. Webala[4], Innocent Twizeyimana[1], Marie-Claire Dusabe[1], Daniel E. Sanchez[2,3], Colin J. Sobek[2,3], Deo Ruhagazi[1], Peace Iribagiza[1], Richard Muvunyi[5], Rodrigo A. Medellin[6]

1 Rwanda Wildlife Conservation Association, Kigali, Rwanda, 2 Bat Ecology & Genetics Lab, School of Forestry, Northern Arizona University, Flagstaff, Arizona, United States of America, 3 The Pathogen and Microbiome Institute, Northern Arizona University, Flagstaff, Arizona, United States of America, 4 Department of Forestry and Wildlife Management, Maasai Mara University, Narok, Kenya, 5 Rwanda Development Board, Kigali, Rwanda, 6 Institute of Ecology, National Autonomous University of Mexico, Mexico City, Mexico

☯ These authors contributed equally to this work.
* Faith.Walker@nau.edu

**Data Availability Statement:** Sequencing data were deposited in NCBI SRA under BioProject ID PRJNA965809, and BioSample accessions are in S1 Table.

## Abstract

Bats are prodigious consumers of agricultural and forest pests, and are, therefore, a natural asset for agricultural productivity, suppressing populations of such pests. This study provides baseline information of diet of 143 bats belonging to eight insectivorous bat species from agricultural areas of Rwanda while evaluating the effectiveness of bats as pest suppressors. Using DNA metabarcoding to analyze bat fecal pellets, 85 different insect species were detected, with 60% ($n = 65$), 64% ($n = 11$) and 78% ($n = 9$) found to be agricultural pests from eastern, northern and western regions, respectively. Given the high percentages of agricultural pests detected, we submit that Rwandan insectivorous bats have the capacity for biocontrol of agricultural pests. Rwandan bat populations should be protected and promoted since they may foster higher crop yields and sustainable livelihoods.

## Introduction

Crop losses due to arthropod pests reduce production of food and cash crops globally [1], and this has become a greater concern due to climate change promoting insect pests [2]. Insectivorous bats prey on a variety of arthropods, many of which are considered major agricultural pests [3,4]. They can consume 30%–100% of their body weight in prey each night [3], and mounting evidence suggests that, through insect consumption, they contribute enormously to both forestry and agriculture [5–10]. In Europe, pest insects have been found to comprise a high percentage of prey diversity and to include pests of high economic impact [11–14]. Charbonnier et al. [13] found that in vineyards in France, bats increased hunting activity when a pest moth species was present. The service of pest suppression has been estimated to have a global value of billions of dollars to agriculture by decreasing insect crop damage and the need

**Funding:** -Rolex Award Collaboration Fund (ON, RAM) https://www.rolex.org/rolex-awards/environment/olivier-nsengimana The funders had no role in the study design, data collection and analysis, decision to publish, or preparation of the manuscript. -Rwanda Wildlife Conservation Association (ON), https://www.rwandawildlife.org/. The funder had a role in data collection and preparation of the manuscript.

**Competing interests:** The authors have declared that no competing interests exist.

to apply pesticides [4,15]. For instance, *Tadarida brasiliensis* bats have an estimated annual value of $741,000 USD as pest control agents on cotton fields in south-central Texas that have a regional value of about six million dollars, a saving of 15% to the industry [16]. This economic pest control value on cotton was based on the bats´ diet of 31% moths [17,18].

The foregoing examples are drawn from North American and European ecosystems because comparable information is largely unavailable for Africa. In Eswatini, *Chaerephon pumilus* and *Mops condylurus* were found selectively foraging over sugarcane plantations from a nearby roost [19], and fed on boring moths (*Eldana saccharina* and *Mythimna phaea*) and stink bugs (*Hemiptera*, *Pentatomidae*), which are major insect pests of sugarcane [20]. In macadamia plantations in South Africa, ecosystem services of insectivorous bats were estimated at $59–139 per hectare through the suppression of stink-bug pest species [21,22]. Finally, in Madagascar, bats in agricultural areas were found to prey on six known pest species [6].

Rwanda is home to >35 insectivorous bat species [23,24]. Although bat density in households was high in eastern Rwanda (42% of 574 assessed houses), very little is known about the potentially important role they play for agriculture. Agriculture represents the second most important contribution to Rwanda´s gross domestic product at 26% [25]. Yet, like in many African countries, bats are shrouded in mystery, misconception, and misinformation, and their ecological roles remain largely unknown. Consequently, they are often feared and persecuted. Much of the recent persecution bats are facing across Rwanda is due to a fear that bats are the source of the virus that causes COVID-19, even though this risk is null [26,27]. The importance of bats especially to agriculture can be demonstrated to communities via outreach supported by research. Studies to determine the economic value of bats is one way to promote bat conservation and to increase the interest of people in protecting them. Increasing local bat populations by decreasing persecution and through artificial roost programs may be an effective and less environmentally destructive means of suppressing insect pests than increasing chemical pesticide use, which may ultimately harm other organisms like insectivorous spiders and birds through poisoning [28,29].

To contribute toward the goal of ultimately showing local communities the ecological services of bats and demonstrating their economic value to policymakers, we collected baseline data on the diet of insectivorous bat species residing in abandoned buildings, residential houses, and caves in the main agricultural areas of Rwanda. We aimed to determine the range of insect pests consumed by bats and therefore assess the effectiveness of these bats as pest suppressors. We hypothesized that a proportion of these insects would be agricultural pests.

## Materials and methods

### Study sites

The study sites were in the eastern part of Rwanda at four main agricultural areas of Kayonza, Nyagatare, Bugesera, and Rwamagana Districts with rice, maize, and tomatoes as the main food crops (Fig 1, Table 1). Eastern regions of Rwanda are affected by prolonged drought while the northern and western regions experience abundant rainfall that at times cause flooding, soil erosion and landslides. The eastern Districts and Bugarama site in the west of Rwanda have low rainfall and hot temperatures. The field activities took place between April 2018 and February 2019. We sampled insectivorous bats across six sites from residential houses, abandoned buildings, and caves. For comparative purposes, additional opportunistic samples were collected from Bugarama (and Musanze Districts, in western and northern Rwanda, respectively).

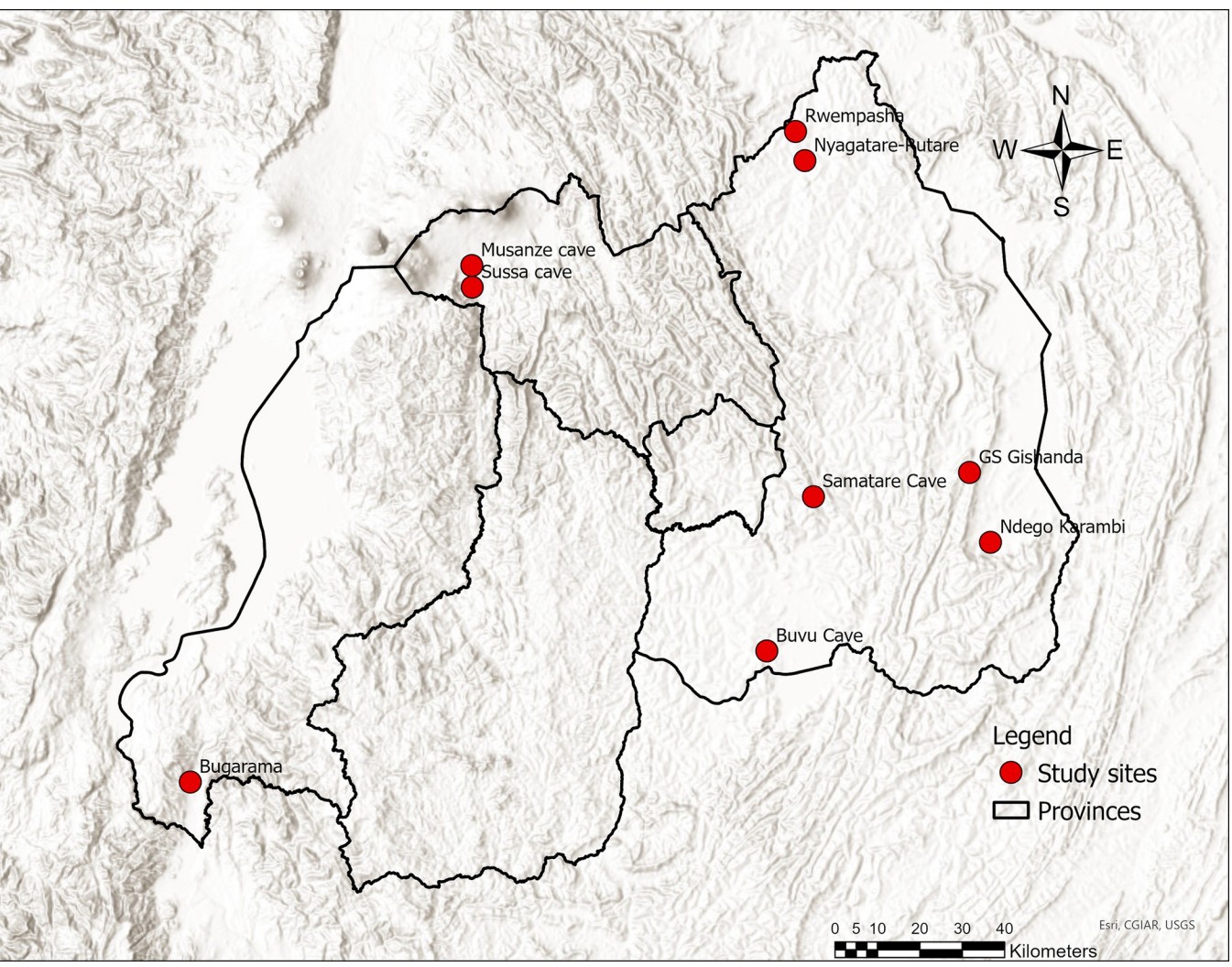

**Fig 1. Map of Rwanda, with study sites (ESRI, CGIAR, USGS).** Layers include ESRI World Topographic Map [30], Province Boundaries [31], and study sites (generated by the authors).

### Field methods

This study was approved by the Rwanda Development Board, Tourism, and Conservation Department (#RDB-T&C/V.U./18) and the Institutional Animal Care and Use Committee at Northern Arizona University (Protocols 07-006-R2, 14–008, and 15–006). All genetic sampling and sequencing complied with relevant guidelines and regulations, and no bats suffered injury or mortality as part of this study. We selected three molossid bat species that are widespread across sub-Saharan Africa and are known to be avid insectivores that consume multiple agricultural pests [20,21,32,33]. Two species, *Chaerephon pumilus* and *Mops condylurus*, are synanthropic, often occupying structures built by humans in large numbers. The third, *Otomops martiensseni*, is fairly ubiquitous and common in caves, forming colonies of up to thousands and is known to consume agricultural pests [21,34].

Bats were captured at roosts in caves or human buildings using mist nets (9 × 2.5 m or 12 × 2.5 m, denier 75/2, mesh 16 × 16 mm, five shelves; Ecotone, Inc., Poland) or hand nets at emergence and return to their roosts. Mist nets were set outside roosts while hand nets were

**Table 1. Sites included in this study, their location in relation to the main agricultural areas/crops, bat species sampled, and estimated bat numbers at the roost sites.**

| District | Site | Crop | Sampled bat species | Estimated bat numbers |
|---|---|---|---|---|
| Kayonza | Ndego house | Tomato plantation | *Chaerephon pumilus* | 4,000 |
| | | | *Mops condylurus* | |
| | Gishanda Primary School | Bramin marshland and Rwinkwavu rice | *Chaerephon pumilus* | 400 |
| | | | *Scotophilus dinganii* | |
| Nyagatare | Rutare cell | Nyagatare rice | *Chaerephon pumilus* | 100 |
| | | | *Mops condylurus* | |
| | Rwempasha conference hall | | *Chaerephon pumilus* | 1,000 |
| Bugesera | Buvu cave | Maize | *Miniopterus minor* | unknown |
| | | | *Hipposideros ruber* | |
| Rwamagana | Samatare Cave | Maize and banana | *Otomops martiensseni* | 300 |
| | | | *Nycteris spp.* | |
| | | | *Afronycteris nana* | |
| Rusizi | Bugarama house | Bugarama rice | *Mops condylurus* | 3,000 |
| | | | *Chaerephon pumilus* | |
| Musanze | Musanze Sussa Cave | Maize | *Otomops martiensseni* | unknown |
| | | | *Hipposideros ruber* | |

used in caves, tunnels, and dwellings to capture bats at roosts. Nets remained open until 0300. Captured bats were placed in clean individual cloth bags (to prevent cross-contamination) and processed a few meters from the capture sites. We measured forearm length of each captured bat with a digital caliper to the nearest 0.1 mm and weight of the bat with Pesola scale to the nearest 0.1 g. Small wing biopsies were also taken for laboratory DNA confirmation of the bat species. Bat identifications and nomenclature followed Patterson & Webala [23].

We collected fecal samples of all captured bats; these accumulated as bats were held in individual cloth bags for an hour on their return from foraging bouts. Additionally, we collected fecal samples from other species that were opportunistically included in the study as a comparison, such as *Miniopterus minor*, *Afronycteris nana*, and *Scotophilus dinganii*, among others. All bat handling techniques and sampling were carried out according to the standards established by the American Society of Mammalogists [35]. We collected approximately 200 fecal pellets from the same bat species into 15 ml sterile conicals, which were then filled with 7.5ml of RNAlater. Biopsy punches for individual bats were collected in 1.5 ml cryovials. To contribute to the reference library for the genetic identification of prey, we collected insects at the study sites at the same time and place as the bats were foraging [36]. We collected insects from tomato (Ndego area), maize (Bugarama area), and rice fields (Nyagatare and Bugarama areas) using UV scorpion light traps. A UV light torch was placed on a string one meter above a white UV-fluorescent cloth to attract insects. We then collected insects with tweezers, each into its own cryovial containing RNAlater. The cryovials with bat fecal samples, wing biopsies, and insects were stored in a dry shipper from the field and then transferred to -80˚ C freezers until transported to Northern Arizona University for DNA analysis.

## Genetic identification of bat diets

**DNA extraction.** We subsampled fecal pellets from RNAlater into 1.5 mL vials and extracted DNA using the QiaAmp Fast DNA Stool Mini Kit (Qiagen, Valencia, CA, USA) following the human DNA analysis protocol. Samples were lysed for 30 min and DNA eluted at 100 μL. We extracted DNA from site-collected insects using the DNeasy Blood and Tissue Kit

(Qiagen, Valencia, CA, USA) following the Animal Tissue protocol. Depending on the size of the specimen, a whole specimen (body length < 7 mm) or a whole leg was initially homogenized with one 5/23" stainless steel grinding ball (OPS Diagnostics) for 40 sec. (6 m/sec) using a FastPrep-24™ 5G Sample Preparation System (MP Biomedicals).

**PCR amplification and sequencing.** We PCR amplified cytochrome c oxidase I (COI) to 1) identify arthropod taxa consumed by the bat species, 2) confirm bat field identification, and 3) identify insects collected from the same localities as the bat species. For goals 1 and 2, we used the ANML primer set (185 bp insert) [37]. For goal 2, we used the Species from Feces (SFF; 202 bp insert) primer set [38,39]. Both primer sets were modified to include 5' universal tails for indexing [40]. Non-template controls (NTCs) were included in all PCR reactions and mock communities were included for Illumina sequencing. The insect mock community included cloned sequences of the ANML target of 24 known arthropod taxa [37,41]. A bat specific mock community included four North American bat species: *Myotis lucifugus*, *Eptesicus fuscus*, *Euderma maculatum*, and *Idionycteris phyllotis*. PCR for both SFF and ANML were run in 15 μL reaction volumes with 3 μL of genomic DNA template. Reactions included 8.46 μL of PCR grade water, 1.5 μL 10X Mg-free PCR buffer (Invitrogen, Thermo Fisher Scientific, Waltham, MA, USA), 1.5 mM MgCl2, 0.2 mM each dNTP, 0.2 μM each primer, 0.16 ug/μL bovine serum albumin (Ambion Ultrapure BSA), and 0.03 U/μL PlatinumTaq DNA polymerase (Invitrogen, Thermo Fisher Scientific). Cycling conditions included an initial denaturation of 94˚C 5 min, 5 cycles of 94˚C for 1 min, 45˚C for 1.5 min, and 72˚C for 1 min, then 35 cycles of 94˚C for 1 min, annealing for 1.5 min (60˚C for SFF and 50˚C for ANML), and 72˚C for 1 min, with a final extension cycle of 72˚C for 5 min. To minimize tag-jumping [42], we used a custom indexing scheme [40] whereby forward and reverse reads were tagged (via PCR) with 8 nucleotide-long indices, extended from the universal tail in a second PCR. An index was used only once per sample. The indexing PCR was run in 25 μL reaction volumes with 2 μL amplicon template, 12.5μ2X Kapa HiFi HotStart ReadyMix (Roche Sequencing, Wilmington, MA, USA), 8.5 μL PCR-grade water, and 1 μL each index primer (starting concentration: 10 uM). Cycling conditions were run at 98˚C for 2 min, followed by 8 cycles of 98˚C for 30 s, 60˚C for 20 s, and 72˚C for 5 min, and then a final extension step of 72˚C for 5 min. We sequenced libraries with Illumina MiSeq V3 600 cycle kits in two sequencing runs, each with 30% PhiX and loaded with 5 μL (3.5 pM) of the pooled libraries. ANML amplicon from site-collected insects was prepared for bi-directional Sanger sequencing using the BigDye Terminator v3.1 kit (Applied Biosystems, Foster City, CA, USA) and sequenced on an ABI3130 Genetic Analyzer (Applied Biosystems, Foster City, CA, USA). We discarded priming regions, verified quality, and edited base calls using Sequencher 5.3 software (http://www.gencodes.com).

**Bioinformatics.** We processed reads for each marker separately. Reads were separated by marker with primers and read-through removed using cutadapt v.2.1 [43]. We used Qiime2 v2020.2 [44] and custom Tidyverse [45] scripts with R statistical software v3.5.1 [46] for all further bioinformatics analysis. Based on quality scores, ANML reads were truncated to 175 bases and SFF reads to 202 bases. We filtered reads by quality, joined paired-ends, denoised, dereplicated into amplicon sequencing variants (ASVs), and filtered chimeras using DADA2 [47]. We further filtered the ASVs of ANML amplicon to exclude bat species [37,41]. ASVs were excluded based on the Vsearch global alignment algorithm [48] if an ASV matched at 95% identity to the bat reference library [49]. We retained any sample that contained > 4000 reads [41]. NTCs did not amplify and so instead of determining a relative abundance threshold using NTCs, we instead used the mock community method [41], indicating that we should exclude ASVs observed in fewer than 0.15% of reads per sample. The ANML and SFF datasets were classified with marker-specific reference libraries [39,41]. ASVs from the ANML dataset were classified with the Vsearch classifier in Qiime2 (classify-consensus-vsearch) at 97%

identity and 89% query coverage. We performed the same classification of the ANML dataset against a reference library of only the site-collected insect sequences under the same parameters. This allowed us to identify whether site-collected insects could be detected in the diet. ASVs from the SFF dataset were classified using the Naïve-Bayes classifier [50] at a 90% confidence threshold. Sequence variants were also cross-referenced using the BOLD identification tool.

We evaluated dietary richness using the ANML ASVs rarified to a minimum read depth of 10,000 reads. Any samples with fewer reads were not included in this analysis. Using base functions in R, we calculated the mean richness per sample and 95% confidence intervals with 10,000 bootstrap iterations. We estimated dietary breadth with a species accumulation curve using the specaccum function (method = "exact") in R package vegan (Dixon 2003). Sanger sequences derived from the site-collected insects were clustered at 98% identity in Vsearch and identified using both Vsearch and the BOLD identification tool [51] to reach consensus. Ambiguous assignments within or among classifiers were collapsed to the lower taxonomic level. The relative proportion of arthropod species in the diet is reported as a percentage of the total number of species detected.

## Results

### Bat fecal sample collection

We obtained fecal samples from 143 bats comprising eight species belonging to five families. Species included *Chaerephon pumilus* (little free-tailed bat; family Molossidae; n = 33), *Mops condylurus* (Angolan free-tailed bat; family Molossidae; n = 23), *Scotophilus dinganii* (African yellow bat; family Vespertilionidae; n = 1), *Miniopterus minor* (least long-fingered bat; family Miniopteridae; n = 20), *Hipposideros ruber* (Noack's roundleaf bat; family Hipposideridae; n = 7), *Otomops martiensseni* (large-eared free-tailed bat; family Molossidae; n = 19), *Afronycteris nana* (banana serotine; family Vespertilionidae; n = 10), and *Nycteris spp* (slit-faced bats; family Nycteridae; n = 4).

### Genetic validation of host species and identification of prey

To validate bat species (SFF marker), we retained 1,030,718 reads for 19/23 samples (mean = 49081.81 ± 37935.62 SD). Two samples failed to amplify and two failed to meet the read minimum threshold, producing only 273 and 1,170 reads. For the ANML marker, we retained 707,652 paired-end reads (mean = 33,697.71 ± 28731.41 SD reads/sample) among 21/23 individuals (324 ASVs). Two of the samples failed to meet the read minimum threshold, producing only 11 and 90 reads. We recovered 25 unique Sanger sequences for the 26 site-collected insects because two specimens shared the same sequence. No negative controls amplified. Genetic identifications also resolved species identifications, and allowed us to determine which pooled samples contained feces from more than one bat species due to mis-identification in the field. One individual field identified to the genus *Nycteris* was genetically identified to a taxonomically unresolved *Nycteris* species known to occur in Sudan (*Nycteris* sp. DMFR-2017).

A taxonomic analysis of arthropod (ANML) sequences revealed that all potential diet items were of the class Insecta. This included 7 orders, 32 families, 40 genera, and 22 species. No negative controls amplified. Although some of the samples revealed a mixture of bat species for the SFF analysis, we still incorporated dietary information from these samples as pooled data to help address our main study question (whether local bat assemblages consumed pests). We found that 24% (n = 25) of unique Sanger sequences generated from site collected arthropods were able to be classified to either genus or species in either Vsearch or the BOLD

**Table 2. Insect specimens that were collected from the study sites and sequenced, along with indication of whether they were identified in the diet of the bat species in this study.**

| Specimen | Taxonomic classification (Vsearch + BOLD Identification Tool) | Detected | No. ASVs | Bat species |
|---|---|---|---|---|
| I001_2018 | Insecta;Lepidoptera | | | |
| I002_2018 | Insecta;Lepidoptera;Noctuidae | | | |
| I002_2019 | Insecta;Lepidoptera | | | |
| I003_2018 | Insecta;Diptera;Culicidae;Mansonia;*Mansonia sp.* | | | |
| I004_2018 | Insecta;Lepidoptera; Noctuidae | | | |
| I006_2018 | Insecta;Diptera;Muscidae | | | |
| I006_2019 | Insecta;Lepidoptera | | | |
| I007_2018 | Insecta;Lepidoptera | | | |
| I008_2018 | Insecta;Diptera;Drosophilidae;Scaptomyza;*Scaptomyza sp.* | | | |
| **I005_2018, I009_2018** | **Insecta;Diptera** | ✓ | 1 | *Chaerephon pumilus* |
| I010_2018 | Insecta;Lepidoptera | | | |
| I010_2019 | Insecta | | | |
| I011_2019 | Insecta;Lepidoptera;Nolidae | | | |
| I012_2019 | Insecta;Lepidoptera;Pyralidae;Emmalocera;*Emmalocera sp.* | | | |
| I013_2018 | Insecta | | | |
| I013_2019 | Insecta;Lepidoptera | | | |
| **I014_2018** | **Insecta;Lepidoptera** | ✓ | 1 | *Hipposideros ruber* |
| I015_2018 | Insecta | | | |
| I016_2019 | Insecta;Lepidoptera | | | |
| I017_2018 | Insecta;Coleoptera;Staphylinidae;Paederus;*Paederus sp.* | | | |
| I018_2018 | Insecta;Lepidoptera;Erebidae;Aroa;*Aroa discalis* | | | |
| **I018_2019** | **Insecta;Hemiptera;Cicadellidae;Cofana;*Cofana spectra*** | ✓ | 1 | *Mops condylurus* |
| **I019_2019** | **Insecta;Diptera;Culicidae; *Culex* or *Lutzia*** | ✓ | 1 | *Chaerephon sp.* |
| I020_2019 | Insecta;Diptera | | | |
| **IO15_2019** | **Insecta;Lepidoptera** | ✓ | 1 | *Mops condylurus, Chaerephon pumilus* |

identification tool (Table 2). Five sequence variants from the fecal data aligned to sequences of five of the site-collected arthropods. However, only two could be classified to a lower taxonomic level than order.

We retained a total of 247 ASVs in the ANML dataset among 18 samples after re-sampling for even read depth of 10,000 reads. The remaining three samples produced fewer than 10,000 reads for ANML and were therefore excluded from analysis of diversity. Although a small number of species were classified, we detected a bootstrapped mean of 15.6 ± 3.3 (95% CI) ASVs (observed SD = 7.37) per sample (minimum = 5, maximum = 28), suggesting that on average we detected 13–19 arthropod taxa per fecal sample. The species accumulation curve (Fig 2) did not approach a clear asymptote, which indicates that the taxonomic richness observed among the individuals was likely a small fraction of the true dietary diversity of the species in this study.

## Arthropod species consumed by insectivorous bats in Rwanda

The diet of insectivorous bats sampled in eastern Rwanda marked 76.4% of insect prey ($n$ = 85) and comprised all orders found in this study except Odonata. Conversely, insect prey of Musanze cave and Bugarama houses constituted 10.5% and 12.9% of the diet of bats in this study, respectively. Among seven orders of insects that were identified in this study, Lepidoptera (Moths) constituted 58.5%, followed by Diptera (14.6%) and Hemiptera (9.8%). The least

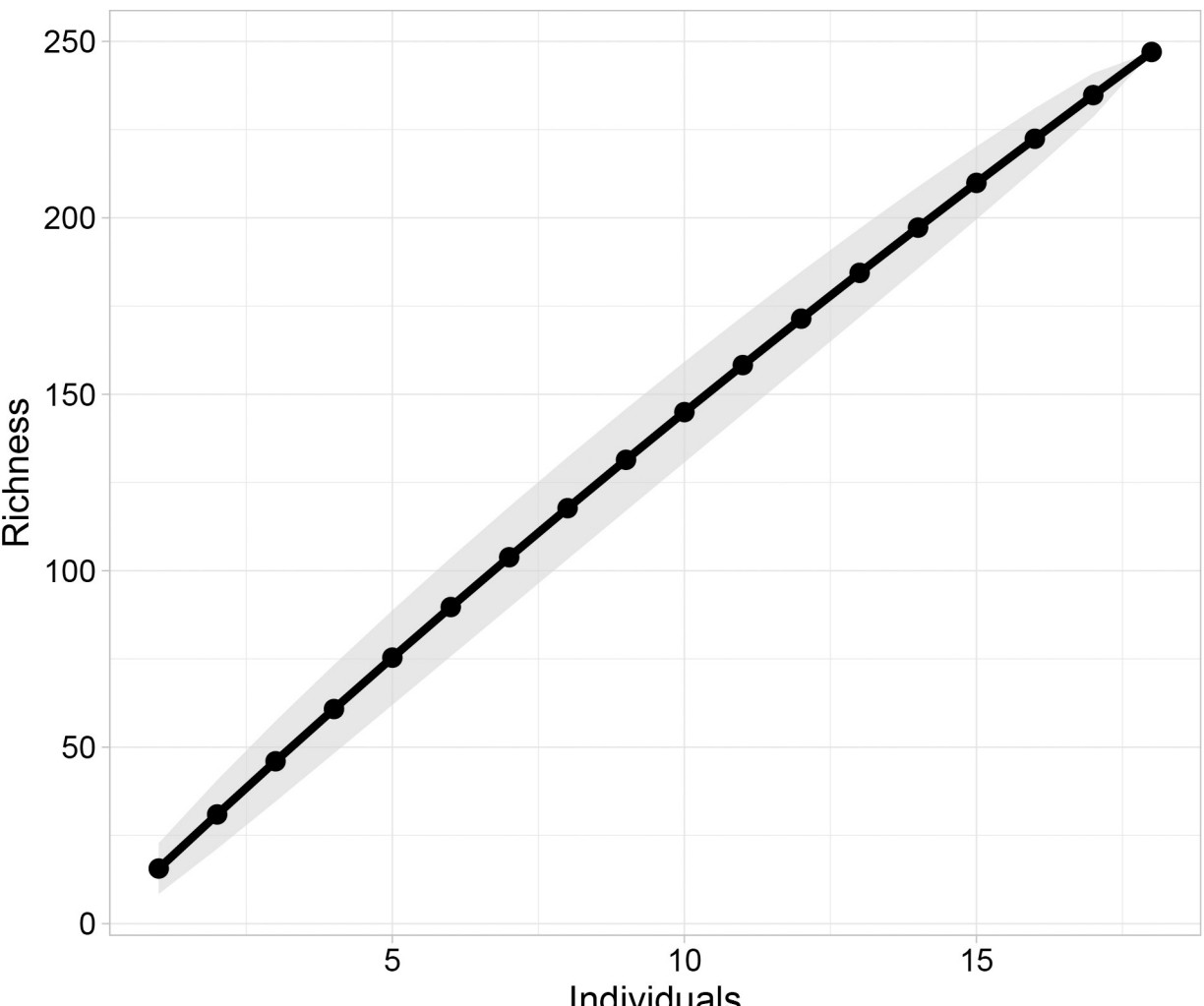

**Fig 2. Species accumulation curve for arthropods detected in bat diets.** The grey shaded region indicates standard error of the estimate.

common orders were Blattodea (4.9%), Coleoptera (4.9%), Orthoptera (4.9%), and Odonata (2.4%) (Fig 3, S1 Table). *Chaerephon pumilus* (n = 33) preyed on all insect orders identified in this study except Odonata. Fecal samples of *C. pumilus* collected in eastern Rwanda consisted of Diptera (40%), Lepidoptera (33.3%), Hemiptera (20%) and Blattodea (6.6%), while the diet of *C. pumilus* from Bugarama area consisted of Diptera (44.4%) (*Culex* and *Drosophila*), Lepidoptera (33.3%), Hemiptera (11.1%) and Odonata (11.1%). *Mops condylurus* (n = 22) consumed only three orders of insects including Lepidoptera (58.3%), Diptera (33.3%) and Hemiptera (8.3%). Fecal pellets of *Hipposideros ruber* (n = 7) and *Miniopterus minor* (n = 20) consisted 100% of Lepidoptera. The diet of *Afronycteris nana* (n = 10) consisted of four insect orders dominated by Diptera and Lepidoptera, marking 37.5% of its total prey for each as well as Hemiptera and Coleoptera (12.5%) per each, respectively. *Nycteris* spp. (n = 1) was found to consume one species *Conocephalus* (Orthoptera) and this was the only species for which we did not find any moth (Lepidoptera) DNA in fecal pellets. *Otomops martiensseni* (n = 19) consumed two orders: Lepidoptera (66.6%) and Diptera (33.3%). *Scotophilus dinganii* (n = 1) consumed Lepidoptera (40%), Coleoptera, Orthoptera and Blattodea comprising 20% of its total

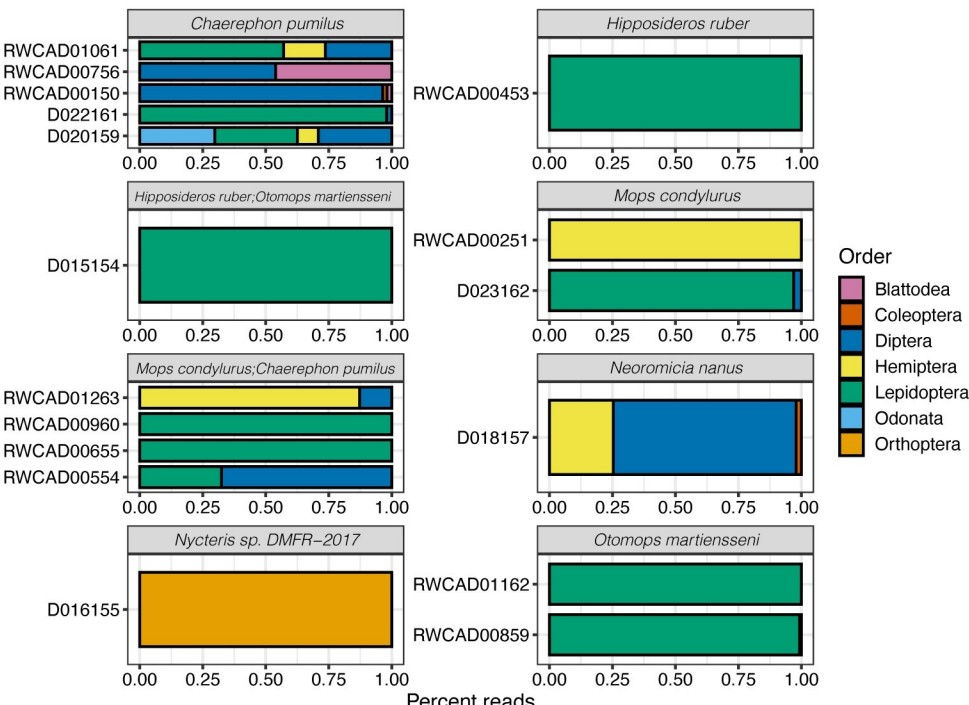

**Fig 3. Sample barplots with aggregated read proportions for insect orders identified in the sample pools.** Only shown are samples with both genetically supported bat species and dietary information (n = 17/23).

diet each. In Bugarama area, *Chaerephon pumilus* (n = 33) was found to consume 10.5% of insect prey of this study, dominated by Diptera (44.4%) (Culex and Drosophila) and were also represented by Lepidoptera (33.3%), Hemiptera (11.1%) and Odonata (11.1%). *Otomops martiensseni* (n = 19) samples from Musanze yielded 12.9% of the prey of this study. These bats consumed almost exclusively Lepidoptera (93%) and some Diptera (7%).

## Pest arthropods in the diet of bats

Sixty percent (n = 65) of insect prey species that were consumed by bats in eastern Rwanda were found to be pests of major agricultural importance. Of the prey species from Musanze caves (northern Rwanda) and in Bugarama (southwestern Rwanda), 63.6% (*n* = 11) and 78% (*n* = 9) of insects were also pest species, respectively. The majority (83.3%) of the pests identified in this study are affiliated with damage to maize, rice, fruits and vegetables (Table 3). In addition, a high number of prey taxa which can potentially become nuisance pests associated with anthropogenic aquatic habitats (e.g., sewage treatment facilities) were found in the diet. These taxa included Dipterans from the family Psychodidae (drain flies) and Chironomidae (Midges) (Table 3). We also found potential disease vectors such as moquitoes (Culicidae), which can transmit various human and livestock diseases. Lepidoptera formed the largest component (56.6%) of the total diet pests (*n* = 30) with Erebidae to be the family most represented (35.4%) among lepidopterans. Geometridae and Crambidae were also common, occurring in 14.7% and 11.7% of the total number of insect prey, respectively. Noctuidae, Sphingidae, and Tortricidae made up 8.8% of the lepidopteran pests each. The least represented families of moth pests were Cosmopterigidae and Nolidae occurring in 5.9% of the total number of pellets per each. Diptera was the second common order made up (23.3%) of the prey pests, with Drosophilidae to be the highest represented (60%), followed by Chironomidae

**Table 3. Prey of human concern detected in the diet of insectivorous bats from Rwanda, as determined by DNA metabarcoding of fecal samples (n = 143) that were pooled into 21 successfully sequenced samples.** A bat species with an asterisk (*) indicates a species only identified in the field. "AND/OR" indicates that the pest was identified in a sample that was mixed with multiple bat species.

| Insect pest | | | | | | | |
|---|---|---|---|---|---|---|---|
| Order | Family | Taxon | Common name | Affiliation | Category | Status & Distribution (Province, this study) | Bat species that consumed pests |
| Hemiptera | Cicadellidae | *Cofana spectra* | White leafhopper | Rice & Maize | Minor in India but major in West Africa | Widespread in West Africa and in India (East) | *Mops condylurus* |
| Lepidoptera | Crambidae | *Duponchelia fovealis* | European pepper moth | strawberries | Major agricultural pest | Worldwide but mostly abundant in Turkey (East) | *Miniopterus minor** |
| Lepidoptera | Erebidae | *Anomis flava* | Cotton looper moth | Cotton | Major agricultural pest | Reported to be severe in Cote d ,Ivoire (East) | *Miniopterus minor** |
| Lepidoptera | Crambidae | *Herpetogramma licarsisalis* | Tropical grass webworms | Pastures | Nuisance pest | Widely distributed, even in Africa (East) | *Hipposideros ruber* |
| Lepidoptera | Erebidae | *Sphingomorpha chlorea* | Sundowner moth, Banana hawk | Fruits | Minor agricultural pest | Widespread in Southeast Asia and Africa (North) | *Otomops martiensseni* AND/OR *Hipposideros ruber* |
| Lepidoptera | Sphingidae | *Daphnis nerii* | Oleander hawk-moth | Plant and forest | Nuisance pest | Minor importance and worldwide distribution (North) | *Otomops martiensseni* |
| Lepidoptera | Tortricidae | *Thaumatotibia leucotreta* | False codling moth | About 70 plants, including crops | Major agricultural pest | Major importance in tropical Africa, Rwanda (East) | *Chaerephon pumilus* |
| Hemiptera | Miridae | *Taylorilygus apicalis* | Brocken backed bug | Hardwood trees, coniferous nurseries | Major agricultural pest | Widely distributed in USA, South Africa, and Guatemala (Kigali) | *Afronycteris nanus* |
| Diptera | Drosophilidae | *Zaprionus indianus* | African big fly | Fruits (Oranges, Peaches and Figs) | Major agricultural pest | Widely distributed in Panama and USA (West) | *Chaerephon pumilus** |
| Lepidoptera | Tortricidae | *Cydia sennae* | - | Fruits | Minor agricultural pest | Known to occur on native fruits of Kenya (East) | *Mops condylurus* |
| Diptera | Drosophilidae | *Drosophila* sp. | Fruit fly | Berries, cherries, grapes, tree fruits | Major fruit pest | Originated in Asia but currently widely distributed (West) | *Chaerephon pumilus* |
| Diptera | Culicidae | *Culex* sp. | Common house mosquitoes | Rift Valley fever virus and West Nile virus | Nuisance pest, disease vector | Widespread in East Africa and known to occur in Rwanda (West) | *Chaerephon pumilus* |
| Lepidoptera | Crambidae | *Parapoynx* sp. | Rice caseworm | Rice | Major agricultural pest | Worldwide (East) | *Miniopterus minor** |
| Lepidoptera | Nolidae | *Earias cupreoviridis* | Cotton green moth | Cotton | Major agricultural pest | Widely distributed in Africa (East) | *Hipposideros ruber* |
| Diptera | Chironomidae | *Kiefferulus brevibucca* | Midge | Aquatic habitats | Major agricultural pest | Widely distributed in Africa (East) | *Chaerephon pumilus* AND/OR *Mops condylurus* |
| Orthoptera | Tettigoniidae | *Conocephalus conocephalus* | African meadow katydid | Rice, sugarcane, forests, fruit orchards | Major agricultural pest | Widely distributed in Asia and Africa (East) | *Nycteris* sp. |

*(Continued)*

**Table 3.** (Continued)

| Insect pest | | | | | | | |
|---|---|---|---|---|---|---|---|
| Order | Family | Taxon | Common name | Affiliation | Category | Status & Distribution (Province, this study) | Bat species that consumed pests |
| Lepidoptera | Noctuidae | *Chrysodeixis* sp. | Golden twin-spot moth | Many fruits, vegetables, ornamental crops | Major agricultural pest | Widely distributed in tropical and subtropical regions (East) | *Hipposideros ruber* AND/OR *Otomops martiensseni* |
| Lepidoptera | Erebidae | *Eudocima* sp. | Fruit piercing moth | Banana, citrus, fig, guava, mango, stonefruit, persimmon, and ripening papaya. | Major and sporadic | East coast of Australia (North) | *Otomops martiensseni* |
| Hemiptera | Delphacidae | *Perkinsiella* sp. | Sugarcane planthopper | Sugarcane | Major agricultural pest | Mostly tropical Asia, Australia, parts of Africa and the Middle East (East) | *Chaerephon pumilus* |
| Diptera | Psychodidae | *Psychoda alternata* | Trickling filter fly or drain fly | Aquatic habitats | Nuisance pest | Originating in North America, has spread around the world (Kigali) | *Afronycteris nanus* |
| Hemiptera | Rhyparochromidae | *Rhyparochromidae* sp. | Seed bug | Strawberries | Major agricultural pest | Worldwide (East) | *Chaerephon pumilus* |
| Lepidoptera | Nolidae | *Nolidae* sp. | Tuft moth | Cotton | Major agricultural pest | Worldwide (Kigali) | *Scotophilus* sp.* |
| Lepidoptera | Geometridae | *Geometridae* sp. | Measuring worm moth, looper, cankerworm inchworm | Apple and tea | Major agricultural pest | Worldwide (Kigali) | *Scotophilus* sp.* AND/OR *Chaerephon pumilus* |
| Hemiptera | Delphacidae | *Delphacidae* sp. | Delphacid planthopper | Rice | Major agricultural pest | Worldwide (East) | *Chaerephon pumilus* |
| Lepidoptera | Noctuidae | *Noctuidae* sp. | Owlet moth, cutworm, or armyworm | More than 80 species of plants | Major agricultural pest | Worldwide (North) | *Miniopterus minor** AND/OR *Otomops martiensseni* |
| Lepidoptera | Erebidae | *Erebidae* sp. | Underwing, litter moth, tiger, lichen, wasp moth, tussock moth, arctic woolly bear moth, and piercing moth | Forest and tea | Major agricultural pest | Found on all continents except Antarctica (North) | All mollosids |
| Lepidoptera | Cosmopterigidae | *Cosmopterigidae* sp. | Cosmet moth | Flowers | Nuisance pest | Widely distributed in Australia and pacific region (East) | *Chaerephon pumilus* |
| Hemiptera | Cicadellidae | *Cicadellidae* sp. | Leafhopper | Maize, potatoes, rice | Major agriculture pest | Worldwide (West) | *Chaerephon pumilus* |

(40%), Psychodidae 20%, and Culicide 20%). Hemiptera (16.6%) was mainly represented by Delphaciidae 40% (an example of *Perkinsiella*), Miridae 20% (*Taylorilygus apicalis)*, Cicadellidae (20%; with *Cofana spectra* accounting all with this family), and Rhyparochromidae 20% (FM). Orthoptera (3.3%) was the least represented order among the pests found in the diet, accounting for only Tettigoniidae (*Conocephalus conocephalus).*

## Discussion

This study used DNA metabarcoding to investigate the diet of insectivorous bats, through which we highlighted their potential ecosystem services associated with pest consumption, especially documenting ingestion of major agricultural pests and mosquitoes. The diet of eight insectivorous bat species comprised two main insect orders, Lepidoptera and Diptera. The presence of mosquitoes (Culicidae) underscores the likelihood of insectivorous bats playing a role in the suppression of certain mosquito species. Clarifying the role of bats in agricultural pest and mosquito vector suppression will likely lead to benefits for both bat conservation and public health. For instance, erecting bat houses to attract bats in farms so that they feed on insects could reduce the use of pesticides, as has been documented in southern Spain [52,53].

The large difference in percentages of insect prey found across the regions could be explained by a sample size effect and/or the different types of habitat sampled. In the east of Rwanda we sampled a larger number of bats in caves, abandoned houses, and human habitations, while in the north and west we had small sample sizes of limited habitat types (16 *Otomops martiensseni* from Musanze cave in the north and 22 *Chaerephon pumilus* from in Bugarama houses in the west).

### Insect diet of eight bat species

Our results are broadly consistent with a study in Cameroon which reported that microbats consumed mainly Lepidoptera, Coleoptera, and Diptera [54] and in Swaziland where two free-tailed bat species, *Chaerephon pumilus* and *Mops condylurus*, also mainly consumed Lepidoptera and Diptera [20]. In this study, bats of the same species living in different regions consumed the same groups of insects, such as *Otomops martiensseni* from Samatare cave (eastern Rwanda) and Musanze cave (northern Rwanda) as well as *Chaerephon pumilus* sampled from the east and west of Rwanda (Bugarama area). Likewise, *C. pumilus* from both areas (east and west of Rwanda) appeared to have consumed the same prey, and prey consisted of similar taxa as in Kenya and South Africa where the most important dietary items were dipterans [33], coleopterans, and hemipterans [55]. The species has also been reported to consume large cockroaches [32,56]. Researchers have highlighted that many insectivorous bats are opportunistic predators [57], choosing particular insect families from different taxa available [58,59], often switching their predatory activity in relation to prey abundance [60]. The diet of bats in our study is in line with what would be expected given that most of the moths are active at night.

The diet of *Mops condylurus* comprised Lepidoptera, Diptera, and Hemiptera. This is consistent with the study by Bohmann et al. [20] who reported the diet of *M. condylurus* to consist mainly of Lepidoptera (46.7%) and Diptera (26.7%), and that the species also consumed a wider range of insect species than *M. condylurus*. We found a similar result: the diet of *C. pumilus* consisted of eight insect orders while the diet of *M. condylurus* consisted of three.

Our results demonstrate that *Hipposideros ruber* and *Miniopterus minor* are moth specialist feeders, as their diet was composed of 100% Lepidoptera. This is consistent with a study that found that *H. ruber* tended to increase foraging around midnight to coincide with moth species such as those of the family Geometridae that have a very late activity peak [61], and other studies that found *H. ruber* to feed predominantly on Lepidoptera [62,63]. In this study we found that *M. minor* fed on five families of Lepidoptera, which is the first information on the diet of the species in Africa.

The diet of *Afronycteris nana* consisted mainly of Diptera and Lepidoptera and to a lesser extent Hemiptera and Coleoptera. These results are consistent with previous studies that found that this species preyed upon Lepidoptera, Coleoptera, and Diptera [64,65]. In South Africa, *A. nana* fed upon Lepidoptera, Diptera, Coleoptera, Hemiptera, Trichoptera, and

Hymenoptera [66]. In the present study, the diet of *Otomops martiensseni* consisted mainly of Lepidoptera with a few dipterans. *O. martiensseni* from Musanze cave, northern Rwanda, consumed Lepidoptera and Diptera and *O. martiensseni* from Samatare cave (eastern Rwanda) consumed mainly Lepidoptera and some Diptera. Our results are consistent with those of another study of 40 droppings collected from *O. martiensseni* in Rwanda, with diet items consisting almost exclusively (97% by volume) of moths [67].

Our single individual *Scotophilus dinganii* consumed mainly Lepidoptera, Coleoptera, Orthoptera and Blattodea, unlike a study in Kenya which reported that the species consumed almost exclusively small beetles [68]. Others have reported that the diet of *S. dinganii* comprises mainly medium-sized Coleoptera, but also may include Hemiptera, Hymenoptera, Isoptera, and Diptera [55,69]. The diet of *Nycteris* sp. in this study included the genus *Conocephalus* (Orthoptera); this was the only species for which we did not detect Lepidoptera in fecal pellets. In Zambia, the diet of *Nycteris macrotis* included Orthoptera, Coleoptera, Isoptera and Diptera [70]. These results point to generalist predator diets for the two bat species within the context of available prey. As generalists, their population dynamics are linked to a wide range of prey [71], with the consumption of a predator per unit time as a function of prey availability [72]. If so, then the differences between our study and other studies could be attributed to prey switching according to the optimal foraging theory [73].

## Pest arthropods in the diet of bats

Globally, insectivorous bats have been reported to regulate insect pests in agricultural systems by decreasing insect crop damage and increasing yield [4,15,16,74]. However, few studies have examined the diversity of agricultural pests and other insects consumed. Our results highlighted that the majority of the total pests are of major agricultural importance, affiliated with damage to maize, rice, fruits, and vegetables. Analysis of the diet also showed nuisance pests associated with anthropogenic aquatic habitats as well as mosquitoes. Among agricultural pests, Lepidoptera was the most frequently reported, which is consistent with a study indicating that most African insectivorous bats feed mainly on Lepidoptera and Coleoptera [55]. Vreysen et al. [75] reported Lepidoptera to be a key insect pest that requires control to avoid losses in many crop systems of temperate, subtropical, and tropical regions of the world. Erebidae was the most common Lepidopteran family found in this study and has been reported to be a pest of several crops as well as forest trees and many ornamental shrubs [76]. Fruit-piercing moths (*Eudocima* sp.) from the family Erebidae are important pests for fruit crops throughout Africa and Southeast Asia [77]. We also found several Geometridae (14.7%) and Crambidae (11.7%). Members of Geometridae are known to cause serious damage to tea plantations (5 to 55% yield loss) [78]. *Duponchelia fovealis* is one of the Crambidae species identified in this study and was reported to cause serious damage to Strawberries in Portugal, France, Italy and Turkey [79–81]. There are records of its occurrence in different regions of Europe, North America, Asia and Africa [79,82,83].

Noctuidae, Sphingidae and Tortricidae were the next most common Lepidopteran families. Noctuidae is one of the top families of concern in agriculture areas. Their larvae are typically known as "cutworms" or "armyworms" due to enormous swarms that destroy crops, orchards, and gardens every year worldwide. In the diet, we found *Noctuid* pest species of major agricultural importance such as *Earias cupreoviridis*, which damaged cotton fields in Tanzania and Uganda [84]. *Anomis flava* is another *Noctuid* pest that we detected in this study and has been reported to damage cotton in Australia, the Philippines, India, and Madagascar [85–87]. *Sphingomorpha chlorea* (a widespread noctuid) was found to cause considerable damage to fruits in Southeast Asia and Africa [88]. Many species of *Chrysodeixis* spp such as *Chrysodeixis*

*chalcites* and *Chrysodeixis eriosoma* feed on a wide variety of fruit and vegetables [89]. *Daphnis nerii* (Sphingidae) reported damage to plants, trees, and shrubs from various families [90]. Another species found in the bat diets was *Thaumatotibia leucotreta* (false codling moth; Tortricidae), reported to damage more than 70 plants and crops in tropical Africa including Rwanda [91]. In South Africa, *T. leucotreta* was reported as one of the most damaging and economically important citrus pests causing the loss of crops equated to more than ZAR 100 Million annually to the South African Citrus Industry [92]. Rice planthoppers (Homoptera: Delphacidae) and leafhoppers (Homoptera: Cicadelidae) reported in this study are important rice pests in rice granary areas [93].

Our results revealed that Dipterans were predominant in bat diets, and 60% were *Drosophila* species. Culicidae (mosquito family) constituted 20% of the Dipteran pests; this suggests that bats are potentially important in the suppression of disease vectors in the Bugarama area and the eastern province of Rwanda. *Chaerephon pumilus* from the two regions fed on prey associated with human habitats such as *Culex* mosquitoes that can carry diseases like the West Nile virus (WNV) [94]. Indeed, the eastern province of Rwanda had the highest seroreactivity of WNV [95]. The area reported a high number of mosquito vectors of this virus, which is transmitted by the bite of Culex mosquitoes where humans and horses are incidental hosts. We also found drain flies (e.g., *Psychoda alternate*) and midges (*Kiefferulus brevibucca*) which are considered nuisance pests when their populations are abundant in residential areas [96]. This justifies the role of insectivorous bats as predators and the ecosystem services they provide expanding beyond their contribution to agricultural-related services.

## Conclusions

This study illustrates the capability for pest suppression and corroborates other studies that show strong ecosystem services provided by bats. It is likely that reference libraries involving African arthropods are depauperate, resulting in the underrepresentation of genus and species-level of prey insects identified in bat fecal samples. In the future, we suggest adding to the reference libraries by collecting and sequencing a large number of insects from the study sites. We also suggest investigating how composition of pest species in bat diets changes over different seasons. Finally, we recommend further research to demonstrate the economic value of bats in order to promote bat conservation and increase the interest of people in protecting bats. This may in turn increase bat population numbers and provide additional ecosystem services.

## Supporting information

**S1 Table. Genetic identifications of bat and insect species in pooled guano samples from Rwanda.**
(XLSX)

## Acknowledgments

RWCA would like to thank Rwanda Development Board, Tourism and Conservation Department for the collaboration on this project and granting research permits. P. Webala is indebted to C. Geiselman for supporting his research on bats of Rwanda. R. Medellin would like to thank Dr. C. Moreno for logistic and technical support. We thank two anonymous reviewers for helpful comments.

## Author Contributions

**Conceptualization:** Olivier Nsengimana, Faith M. Walker, Paul W. Webala, Richard Muvunyi, Rodrigo A. Medellin.

**Data curation:** Innocent Twizeyimana, Marie-Claire Dusabe, Daniel E. Sanchez, Colin J. Sobek.

**Formal analysis:** Innocent Twizeyimana, Marie-Claire Dusabe, Daniel E. Sanchez, Deo Ruhagazi, Peace Iribagiza.

**Funding acquisition:** Olivier Nsengimana, Rodrigo A. Medellin.

**Investigation:** Olivier Nsengimana, Faith M. Walker, Paul W. Webala, Innocent Twizeyimana, Marie-Claire Dusabe, Daniel E. Sanchez, Colin J. Sobek, Rodrigo A. Medellin.

**Methodology:** Faith M. Walker, Paul W. Webala, Innocent Twizeyimana, Marie-Claire Dusabe.

**Project administration:** Olivier Nsengimana, Faith M. Walker, Rodrigo A. Medellin.

**Resources:** Olivier Nsengimana.

**Software:** Daniel E. Sanchez.

**Supervision:** Olivier Nsengimana, Faith M. Walker, Rodrigo A. Medellin.

**Validation:** Daniel E. Sanchez, Colin J. Sobek.

**Visualization:** Daniel E. Sanchez.

**Writing – original draft:** Faith M. Walker, Paul W. Webala, Innocent Twizeyimana, Marie-Claire Dusabe, Daniel E. Sanchez, Rodrigo A. Medellin.

**Writing – review & editing:** Olivier Nsengimana, Faith M. Walker, Paul W. Webala, Innocent Twizeyimana, Daniel E. Sanchez, Colin J. Sobek, Deo Ruhagazi, Peace Iribagiza, Richard Muvunyi, Rodrigo A. Medellin.

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
