## [Decision Letter · Decision Letter 0]

22 Mar 2023

PONE-D-22-33514Our good neighbors: Understanding ecosystem services provided by insectivorous bats in RwandaPLOS ONE

Dear Dr. Walker,

Thank you for submitting your manuscript to PLOS ONE. After careful consideration, we feel that it has merit but does not fully meet PLOS ONE’s publication criteria as it currently stands. Therefore, we invite you to submit a revised version of the manuscript that addresses the points raised during the review process. Please submit your revised manuscript by May 06 2023 11:59PM. If you will need more time than this to complete your revisions, please reply to this message or contact the journal office at plosone@plos.org. Please include the following items when submitting your revised manuscript:A rebuttal letter that responds to each point raised by the academic editor and reviewer(s). You should upload this letter as a separate file labeled 'Response to Reviewers'.A marked-up copy of your manuscript that highlights changes made to the original version. You should upload this as a separate file labeled 'Revised Manuscript with Track Changes'.An unmarked version of your revised paper without tracked changes. You should upload this as a separate file labeled 'Manuscript'.If applicable, we recommend that you deposit your laboratory protocols in protocols.io to enhance the reproducibility of your results. Protocols.io assigns your protocol its own identifier (DOI) so that it can be cited independently in the future. For instructions see: https://journals.plos.org/plosone/s/submission-guidelines#loc-laboratory-protocols. Additionally, PLOS ONE offers an option for publishing peer-reviewed Lab Protocol articles, which describe protocols hosted on protocols.io. Read more information on sharing protocols at https://plos.org/protocols?utm_medium=editorial-email&utm_source=authorletters&utm_campaign=protocols.

We look forward to receiving your revised manuscript. With our apologies for the long reviewing process.

Kind regards,

Camille Lebarbenchon

Academic Editor

PLOS ONE

Journal Requirements:

4. We note that Figure 1 in your submission contain map/satellite image which may be copyrighted. All PLOS content is published under the Creative Commons Attribution License (CC BY 4.0), which means that the manuscript, images, and Supporting Information files will be freely available online, and any third party is permitted to access, download, copy, distribute, and use these materials in any way, even commercially, with proper attribution. For these reasons, we cannot publish previously copyrighted maps or satellite images created using proprietary data, such as Google software (Google Maps, Street View, and Earth). For more information, see our copyright guidelines: http://journals.plos.org/plosone/s/licenses-and-copyright.

Reviewers' comments:

Reviewer's Responses to Questions

**Comments to the Author**

1. Is the manuscript technically sound, and do the data support the conclusions?

Reviewer #1: Partly

Reviewer #2: Partly

2. Has the statistical analysis been performed appropriately and rigorously? 

Reviewer #1: Yes

Reviewer #2: Yes

3. Have the authors made all data underlying the findings in their manuscript fully available?

Reviewer #1: Yes

Reviewer #2: Yes

4. Is the manuscript presented in an intelligible fashion and written in standard English?

Reviewer #1: Yes

Reviewer #2: Yes

5. Review Comments to the Author

Reviewer #1: The authors investigated the diet of several insectivorous bat species in Rwanda, the presence of pest species and the potential role of molossid bats as pest suppressors. I really enjoyed the manuscript and I believe it would make a great contribution to the bat conservation research. However, I do think that some aspects must be improved/clarified before publication (technical replicates, inclusion of contaminated samples in the analyses). Please find my major and minor comments below.

L64 : Please, precise what GDP stands for.

Table 1: It’s a tiny detail but it seems that the table is cropped a little bit on the right side

L151: typo, space is missing between “1” and “and”

L148: I understand that you’re are limited in page number, but maybe provided a little bit more information here (in my understanding you did a two step PCR but that’s something you can precise here for example, as well as amplicon size for each primer pair etc)

L152: Following which PCR conditions? The same than for ANML? (amplicon size etc)

153: What about the technical replicates? If you used technical replicates, please indicate how many and how the data were processed accordingly. If not, please justify why as it is highly highly recommended in metabarcoding study to overcome PCR and sequencing stochasticity.

L160: For each run, please indicate the volume of library that was loaded & %PhiX. Please also indicate whether one run was done for ANML and one for SFF or whether they were mixed. Also, please indicate how many raw reads you obtained for each run in the results.

L166: Any filtering using the NTC or technical replicates? If not, please justify your choice.

L173: I don’t understand, why did you exclude ASV of bat species? It would have been interesting to compare the ANML & SFF IDs, especially as you got mixed bat ID within samples.

L176: I’m sorry I don’t understand how the mock communities informed to set 4000 reads as a threshold, could you elaborate a little bit more on that please?

L183: Why using a different classifier approach?

L186: Please precise the tool (function/package) that you used for the calculation.

L192: Any check on geographic ranges (e.g. species not occurring in Rwanda or Africa)?

L196 : I wonder whether a quick reminder of the digestion time would be useful as you are measuring before/after (here or even in the Discussion).

L200/201 : Is the « : » missing? (like 1830 should be 18 :30 ?)

L224 : Why only 19, did the four remaining samples not amplify (SFF marker) or did they have less than 4000 reads?

L226: Same question, why 21/23 (although I counted 22 samples in your Suppl. table - maybe double check the numbers and percentages)?

L230: Please add that there are six samples showing a mix of different bat species (RWCAD00554, 655, 960, 15154, 14153, 1263). I strongly recommend discarding them when doing analysis per bat species, even when the number of reads of the second bat specie is ~low (<100) because it is impossible to know if some prey belong to one or another bat species.

L234: Be consistent with sample ID (RWCA00251, RWCAD00655 etc). RWCAD00655 identification mismatch could be due to contamination with the other bat species. Also, I would not mention this sample here as you indicated a few lines above that you have discarded it.

L241: Why giving more details on Molossids and not on the other families?

L250: Why these 18 individuals? Is the previous paragraph based on these 18 individuals? If so, please clarify.

L258: I would suggest reminding the number of samples per species when describing their diet, ex. “Nycteris spp. (n=1) was found to consume (…) “ . Something to keep in mind when discussing the results in the Discussion.

L264 : typo, Blattodea

L276: typo, “%” is missing for Odonata

L347-350 : results section?

L388: I wonder whether you could add Order/family of the pest species to the Table 3. I do think it would be very helpful to make the link with the text that describes per family and order.

Global comment on Discussion: In my opinion some parts of the Discussion are too repetitive with the Results. Your data are very interesting, but sometimes it feels like you could go a little further (e.g. L380 to L387: what could explain the differences between your study and the other studies?). Also, sample size is (very) limited for some species, it would be relevant to mention it in the Discussion.

Literature: Check reference format. Sometimes info is missing (ex #10 volume/pages) and species name should be in italic.

Fig2: Please add in the caption what the grey shaded area represents.

Fig3: Why only 17 samples are shown?

Fig4: Please, precise the unit of the Y axis (grammes). Why is only the genus shown? This analysis is by species, isn’t it?

Table 3: It would be great to have another column showing in which area (North, East, West, South) they were detected in your study to make the link with the Discussion.

Supplementary tables: Please, add a caption above both tables. Table Final results bats: Please check and homogeneize bat species name: Hipposideros ruber or Hyposiderus ruber; Rhinolophus not Rhinolofus etc. Why is Visual ID at the genus level in 2019?

“Ambiguous_taxa”? Please clarify (should be the lowest common shared assignment i.e. Chrysomya?) [sample D014153]

Reviewer #2: This is a valuable manuscript that investigates the diet of bat species in Rwanda and demonstrates, through molecular diet analysis and insect sampling in the field, that these bats consume several agricultural pests. The authors are to be commended for addressing a timely topic in a region where it is understudied compared to other areas. However, there is room for improvement, as outlined below.

1. Scientific articles should frame hypotheses and predictions rather than research questions. I urge the authors to reframe the objectives of their study by presenting well-thought-out hypotheses and measurable predictions. While the research questions are clear, reframing them in this way should not require significant effort.

2. In the introduction, better coverage of European studies would increase the international appeal of the manuscript. Recent articles from France, Italy, Portugal, and the UK cover the topic and should be included.

3. In many sections of the manuscript, the authors imply that bats "control" pests, but the term control implies a demographic effect on the prey or visible reduced damage to the cultivation. I recommend using more neutral wording such as suppression or consumption unless studies providing strong evidence of control are mentioned.

4. The authors' method of estimating the amount of prey consumed by measuring the weight of bats leaving and returning to the roost after a hunting night is not reliable. To make this approach more dependable, the authors should have used a repeated-measures analysis, capturing the same bat twice in the two temporal phases to determine the weight difference as the amount of food eaten. Since this was not done, I suggest dropping this section entirely. The manuscript already contains a wealth of valuable and interesting information. Also, please refrain from discussing non-significant results like they were significant. P>0.05 means there is no difference.

Minor comments:

1) In line 68, the authors state that the risk of bats carrying and transmitting SARS-CoV2 is "low to non-existent”, but in fact it is null – please correct

2) Line 114: clarification is needed on whether the nets were kept open for three hours.

3) Table 2: Hipposideros is misspelt.

4) Wordiness can be reduced for better readability, e.g., "Chaerephon pumilus was found to prey on all insect orders” would become “ Chaerephon pumilus preyed on all insect orders

5) Line 306: clarification is needed on how many bats were used to make this estimate (but please see my major point on this part of the study).

6) In the discussion, the authors should avoid presenting results and instead focus on discussing them. Percentages of prey occurrence should not be repeated here, they are already shown in the results.

7) In line 413, it should be “Noctuid”

6. PLOS authors have the option to publish the peer review history of their article (what does this mean?). If published, this will include your full peer review and any attached files.

Reviewer #1: No

Reviewer #2: No

---

## [Author Response · Author response to Decision Letter 0]

22 May 2023

Journal Requirements:

1. Please ensure that your manuscript meets PLOS ONE's style requirements, including those for

file naming. The PLOS ONE style templates can be found at

and

https://journals.plos.org/plosone/s/file?id=ba62/PLOSOne_formatting_sample_title_authors_aff

iliations.pdf

We have ensured that the manuscript meets PLOS ONE’s style requirements.

2. In your Data Availability statement, you have not specified where the minimal data set

underlying the results described in your manuscript can be found. PLOS defines a study's

minimal data set as the underlying data used to reach the conclusions drawn in the manuscript

and any additional data required to replicate the reported study findings in their entirety. All

PLOS journals require that the minimal data set be made fully available. For more information

about our data policy, please see http://journals.plos.org/plosone/s/data-availability.

The data are in NCBI Sequence Read Archive BioProject ID PRJNA965809.

Upon re-submitting your revised manuscript, please upload your study’s minimal underlying

data set as either Supporting Information files or to a stable, public repository and include the

relevant URLs, DOIs, or accession numbers within your revised cover letter. For a list of

acceptable repositories, please see http://journals.plos.org/plosone/s/data-availability#locrecommended-repositories. Any potentially identifying patient information must be fully

anonymized.

Important: If there are ethical or legal restrictions to sharing your data publicly, please explain

these restrictions in detail. Please see our guidelines for more information on what we consider

unacceptable restrictions to publicly sharing data: http://journals.plos.org/plosone/s/dataavailability#loc-unacceptable-data-access-restrictions. Note that it is not acceptable for the

authors to be the sole named individuals responsible for ensuring data access.

We will update your Data Availability statement to reflect the information you provide in your

cover letter.

3. We note that you have stated that you will provide repository information for your data at

acceptance. Should your manuscript be accepted for publication, we will hold it until you

provide the relevant accession numbers or DOIs necessary to access your data. If you wish to

make changes to your Data Availability statement, please describe these changes in your cover

letter and we will update your Data Availability statement to reflect the information you provide.

The data are in NCBI Sequence Read Archive BioProject ID PRJNA965809

4. We note that Figure 1 in your submission contain map/satellite image which may be

copyrighted. All PLOS content is published under the Creative Commons Attribution License

(CC BY 4.0), which means that the manuscript, images, and Supporting Information files will be

freely available online, and any third party is permitted to access, download, copy, distribute,

and use these materials in any way, even commercially, with proper attribution. For these

reasons, we cannot publish previously copyrighted maps or satellite images created using

proprietary data, such as Google software (Google Maps, Street View, and Earth). For more

information, see our copyright guidelines: http://journals.plos.org/plosone/s/licenses-andcopyright.

We require you to either (1) present written permission from the copyright holder to publish

these figures specifically under the CC BY 4.0 license, or (2) remove the figures from your

submission.

Figure 1 was created with ArcGIS, by ESRI, which allows publication with attribution. We have

placed attribution on the map and in the caption.

Reviewer 1

The authors investigated the diet of several insectivorous bat species in Rwanda, the presence of

pest species and the potential role of molossid bats as pest suppressors. I really enjoyed the

manuscript and I believe it would make a great contribution to the bat conservation research.

However, I do think that some aspects must be improved/clarified before publication (technical

replicates, inclusion of contaminated samples in the analyses). Please find my major and minor

comments below.

L64 : Please, precise what GDP stands for.

We have defined GDP.

Table 1: It’s a tiny detail but it seems that the table is cropped a little bit on the right side

Rectified.

L151: typo, space is missing between “1” and “and”

We have added a space.

L148: I understand that you’re are limited in page number, but maybe provided a little bit more

information here (in my understanding you did a two step PCR but that’s something you can

precise here for example, as well as amplicon size for each primer pair etc)

We included conditions for SFF and ANML PCR, and indexing PCR. We also specified the

insert size for both markers.

L152: Following which PCR conditions? The same than for ANML? (amplicon size etc)

We included all PCR conditions and noted insert size.

153: What about the technical replicates? If you used technical replicates, please indicate how

many and how the data were processed accordingly. If not, please justify why as it is highly

highly recommended in metabarcoding study to overcome PCR and sequencing stochasticity.

We did not use technical replicates. We were interested in the taxa detected, not in read

proportions, which is particularly subject to stochastic processes. Mock communities were used

as positive controls; all species in these mock communities were detected.

L160: For each run, please indicate the volume of library that was loaded & %PhiX. Please also

indicate whether one run was done for ANML and one for SFF or whether they were mixed.

Also, please indicate how many raw reads you obtained for each run in the results.

This is good that the reviewer is asking for this because it is not reported enough in the

metabarcoding literature and will determine whether data is clean or dirty. We have clarified

PhiX%. According to the sequencing core we used, all libraries were loaded at 5 uL. However,

they feel that the concentration of the libraries is important for optimal cluster density for low

diversity libraries such as ours. Therefore, we also reported the final loading concentration.

L166: Any filtering using the NTC or technical replicates? If not, please justify your choice.

Our NTCs did not amplify and we did not use technical replicates. We used O’Rourke et al.’s

(2020) mock community method to infer filtering thresholds. We have edited to improve clarity.

L173: I don’t understand, why did you exclude ASV of bat species? It would have been

interesting to compare the ANML & SFF IDs, especially as you got mixed bat ID within samples.

This is a great idea. However, respectfully, SFF primers were designed for bat identification and

are much more appropriate for bat species ID than ANML, which was designed to more

efficiently amplify arthropods. We believe this comparison does not help us address our main

questions and would be more appropriately addressed with an ad hoc study design. Based on

other experiments, we have anecdotal evidence that ANML is quite inconsistent at recovering bat

species and that SFF recovers more.

L176: I’m sorry I don’t understand how the mock communities informed to set 4000 reads as a

threshold, could you elaborate a little bit more on that please?

This was a miscommunication. The mock community did not inform the setting of 4000 read

threshold. It helped us only infer a relative abundance threshold. We have edited for clarity.

L183: Why using a different classifier approach?

We used different classifiers for SFF and ANML markers based on validation studies for each

marker. Naïve Bayes classification outperforms alignment-based classification for SFF (Walker

et al. 2016), whereas Vsearch global-alignment-based classification provides better species level

resolution for arthropods.

L186: Please precise the tool (function/package) that you used for the calculation.

We used base functions in R and did not require the use of an external package. We clarified in

text.

L192: Any check on geographic ranges (e.g. species not occurring in Rwanda or Africa)?

Yes, we checked for geographic ranges for detected bats and arthropods.

L196 : I wonder whether a quick reminder of the digestion time would be useful as you are

measuring before/after (here or even in the Discussion).

We have deleted this section, as suggested by Reviewer 2.

L200/201 : Is the « : » missing? (like 1830 should be 18 :30 ?)

A semicolon is not used for time in this format.

L224 : Why only 19, did the four remaining samples not amplify (SFF marker) or did they have

less than 4000 reads?

Two samples did not amplify and two were omitted because they failed to meet the read

minimum threshold (they only produced 273 and 1170 reads). We included this information in

this section.

L226: Same question, why 21/23 (although I counted 22 samples in your Suppl. table - maybe

double check the numbers and percentages)?

We clarified that these two ANML samples did not meet the read minimum requirements,

producing only 11 and 1,170 reads, and were therefore excluded. The first tab in the S1 table has

22 samples because some samples only amplified for bats and not arthropods and vice versa. One

sample failed for both markers and so the 23rd sample is not included in the taxonomy summary.

L230: Please add that there are six samples showing a mix of different bat species

(RWCAD00554, 655, 960, 15154, 14153, 1263). I strongly recommend discarding them when

doing analysis per bat species, even when the number of reads of the second bat specie is ~low

(<100) because it is impossible to know if some prey belong to one or another bat species.

We agree with this recommendation. However, our study question is community level (do local

bat assemblages consume pest species?) and strongly believe the dietary information is still

informative to our question. Excluding these samples would be more appropriate for an explicit

dietary comparison among species and this was not an objective of this study. Our question does

not require knowing which bat species consumed a pest, although we do agree that a more pure

sample set would have been more ideal. We have added a sentence to clarify this decision.

L234: Be consistent with sample ID (RWCA00251, RWCAD00655 etc). RWCAD00655

identification mismatch could be due to contamination with the other bat species. Also, I would

not mention this sample here as you indicated a few lines above that you have discarded it.

We have corrected the sample names for consistency and omitted RWCA00655 from this line.

L241: Why giving more details on Molossids and not on the other families?

We removed this detail from a previous version of this manuscript.

L250: Why these 18 individuals? Is the previous paragraph based on these 18 individuals? If so,

please clarify.

We forgot to include that we rarified the ASV table for an even sampling depth of 10,000 reads.

Any sample with fewer reads would have been discarded. We added this information to the

methods and clarified in the results.

L258: I would suggest reminding the number of samples per species when describing their diet,

ex. “Nycteris spp. (n=1) was found to consume (…) “ . Something to keep in mind when

discussing the results in the Discussion.

We have added the number of individuals for each species.

L264 : typo, Blattodea

We have corrected this typo.

L276: typo, “%” is missing for Odonata

We have added the % sign.

L347-350 : results section?

We have moved this to the Results.

L388: I wonder whether you could add Order/family of the pest species to the Table 3. I do think

it would be very helpful to make the link with the text that describes per family and order.

We added order and family to the table.

Global comment on Discussion: In my opinion some parts of the Discussion are too repetitive

with the Results. Your data are very interesting, but sometimes it feels like you could go a little

further (e.g. L380 to L387: what could explain the differences between your study and the other

studies?). Also, sample size is (very) limited for some species, it would be relevant to mention it

in the Discussion.

We have made the Discussion less repetitive with the Results.

Literature: Check reference format. Sometimes info is missing (ex #10 volume/pages) and

species name should be in italic.

We have checked the reference format. Some papers, such as #10, are recently published and

have not been assigned volume/pages yet.

Fig2: Please add in the caption what the grey shaded area represents.

We have clarified that this is the standard error of the estimate.

Fig3: Why only 17 samples are shown?

We only included samples that had genetic IDs and dietary information. Given that we identified

mislabeled, mis-identified, or contaminated samples, we felt it could be misleading to include

visual IDs. We have edited the caption for clarity.

Fig4: Please, precise the unit of the Y axis (grammes). Why is only the genus shown? This

analysis is by species, isn’t it?

As per the 2nd reviewer’s suggestion, we removed this analysis and this associated figure from

our manuscript.

Table 3: It would be great to have another column showing in which area (North, East, West,

South) they were detected in your study to make the link with the Discussion.

We have added this information to Table 3.

Supplementary tables: Please, add a caption above both tables.

We have added captions.

Table Final results bats: Please check and homogeneize bat species name: Hipposideros ruber

or Hyposiderus ruber; Rhinolophus not Rhinolofus etc.

We have corrected and homogenized bat species names.

Why is Visual ID at the genus level in 2019?

In 2019, visual ID was limited to the genus level due to the absence of the expert Dr. Paul

Webala, who would help with identification confirmation. We were confident enough to identify

up to the genus level. We took morphological measurements of all bats captured, and took

photos of bats and sent them to Dr. Paul Webala for further identification. We decided to present

the data as recorded in the field because we knew there would be additional DNA analysis to

confirm bat identifications.

“Ambiguous_taxa”? Please clarify (should be the lowest common shared assignment i.e.

Chrysomya?) [sample D014153].

We have collapsed to genus.

Reviewer 2

This is a valuable manuscript that investigates the diet of bat species in Rwanda and

demonstrates, through molecular diet analysis and insect sampling in the field, that these bats

consume several agricultural pests. The authors are to be commended for addressing a timely

topic in a region where it is understudied compared to other areas. However, there is room for

improvement, as outlined below.

1. Scientific articles should frame hypotheses and predictions rather than research questions. I

urge the authors to reframe the objectives of their study by presenting well-thought-out

hypotheses and measurable predictions. While the research questions are clear, reframing them

in this way should not require significant effort.

We have recast the questions as hypotheses.

2. In the introduction, better coverage of European studies would increase the international

appeal of the manuscript. Recent articles from France, Italy, Portugal, and the UK cover the

topic and should be included.

We have increased the coverage of European studies in the Introduction.

3. In many sections of the manuscript, the authors imply that bats "control" pests, but the term

control implies a demographic effect on the prey or visible reduced damage to the cultivation. I

recommend using more neutral wording such as suppression or consumption unless studies

providing strong evidence of control are mentioned.

We have removed the term “control” from our study.

4. The authors' method of estimating the amount of prey consumed by measuring the weight of

bats leaving and returning to the roost after a hunting night is not reliable. To make this

approach more dependable, the authors should have used a repeated-measures analysis,

capturing the same bat twice in the two temporal phases to determine the weight difference as

the amount of food eaten. Since this was not done, I suggest dropping this section entirely. The

manuscript already contains a wealth of valuable and interesting information. Also, please

refrain from discussing non-significant results like they were significant. P>0.05 means there is

no difference.

We have deleted this section.

Minor comments:

1) In line 68, the authors state that the risk of bats carrying and transmitting SARS-CoV2 is "low

to non-existent”, but in fact it is null – please correct

We have corrected this statement.

2) Line 114: clarification is needed on whether the nets were kept open for three hours.

This is a typo. Nets were open during bat emergence for foraging. When no bats were still

emerging, the nets were closed. The nets were reopened when the bats were returning back to the

roost, with the majority of bats returning at 3 am.

3) Table 2: Hipposideros is misspelt.

We have corrected this misspelling.

4) Wordiness can be reduced for better readability, e.g., "Chaerephon pumilus was found to prey

on all insect orders” would become “ Chaerephon pumilus preyed on all insect orders

We have reduced wordiness for better readability.

5) Line 306: clarification is needed on how many bats were used to make this estimate (but

please see my major point on this part of the study).

We have deleted this section.

6) In the discussion, the authors should avoid presenting results and instead focus on discussing

them. Percentages of prey occurrence should not be repeated here, they are already shown in the

results.

We have removed repeated results in the Discussion.

7) In line 413, it should be “Noctuid”

We have italicized Noctuid

---

## [Editor Report · Decision Letter 1]

7 Jun 2023

Our good neighbors: Understanding ecosystem services provided by insectivorous bats in Rwanda

PONE-D-22-33514R1

Dear Dr. Walker,

We’re pleased to inform you that your manuscript has been judged scientifically suitable for publication and will be formally accepted for publication once it meets all outstanding technical requirements.

Kind regards,

Camille Lebarbenchon

Academic Editor

PLOS ONE

---

## [Editor Report · Acceptance letter]

15 Jun 2023

PONE-D-22-33514R1 

Our good neighbors: Understanding ecosystem services provided by insectivorous bats in Rwanda 

Dear Dr. Walker:

I'm pleased to inform you that your manuscript has been deemed suitable for publication in PLOS ONE. Congratulations! Your manuscript is now with our production department. 

Kind regards, 

on behalf of

Dr. Camille Lebarbenchon 

Academic Editor

PLOS ONE